# OPTION DISCOVERY USING DEEP SKILL CHAINING

**Akhil Bagaria**
Department of Computer Science
Brown University
Providence, RI, USA
`akhil_bagaria@brown.edu`

**George Konidaris**
Department of Computer Science
Brown University
Providence, RI, USA
`gdk@brown.edu`

## ABSTRACT

Autonomously discovering temporally extended actions, or *skills*, is a longstanding goal of hierarchical reinforcement learning. We propose a new algorithm that combines skill chaining with deep neural networks to autonomously discover skills in high-dimensional, continuous domains. The resulting algorithm, *deep skill chaining*, constructs skills with the property that executing one enables the agent to execute another. We demonstrate that deep skill chaining significantly outperforms both non-hierarchical agents and other state-of-the-art skill discovery techniques in challenging continuous control tasks.[1] [2]

## 1 INTRODUCTION

Hierarchical reinforcement learning (Barto & Mahadevan, 2003) is a promising approach for solving long-horizon sequential decision making problems. Hierarchical methods lower the decision making burden on the agent through the use of problem specific action abstractions (Konidaris, 2019). While the use of temporally extended actions, or *options* (Sutton et al., 1999), has been shown to accelerate learning (McGovern & Sutton, 1998), there remains the question of skill discovery: how can agents autonomously construct useful skills via interaction with the environment? While a large body of work has sought to answer this question in small discrete domains, skill discovery in high-dimensional continuous spaces remains an open problem.

An early approach to skill discovery in continuous-state environments was skill chaining (Konidaris & Barto, 2009b), where an agent constructs a sequence of options that target a salient event in the MDP (for example, the goal state). The skills are constructed so that successful execution of each option in the chain allows the agent to execute another option, which brings it closer still to its eventual goal. While skill chaining was capable of discovering skills in continuous state spaces, it could only be applied to relatively low-dimensional state-spaces with discrete actions.

We introduce a new algorithm that combines the core insights of skill chaining with recent advances in using non-linear function approximation in reinforcement learning. The new algorithm, *deep skill chaining*, scales to high-dimensional problems with continuous state and action spaces. Through a series of experiments on five challenging domains in the MuJoCo physics simulator (Todorov et al., 2012), we show that deep skill chaining can solve tasks that otherwise cannot be solved by non-hierarchical agents in a reasonable amount of time. Furthermore, the new algorithm outperforms state-of-the-art deep skill discovery algorithms (Bacon et al., 2017; Levy et al., 2019) in these tasks.

## 2 BACKGROUND AND RELATED WORK

Sequential decision making problems can be formalized as Markov Decision Processes (MDPs). We consider goal-oriented episodic MDPs, where $S$ denotes the state space, $A$ is the action space, $R$ is the reward function, $\mathcal{T}$ is the transition function, $\gamma$ is the discount factor and $g \in S$ is the terminating goal state (Sutton & Barto, 2018). Unlike goal-conditioned algorithms (Sutton et al., 2011; Schaul et al., 2015), we do not require that $g$ be known; instead we assume access to an indicator function $\mathbb{1}_g : s \in S \to \{0, 1\}$ which the agent can query to determine if it has reached the MDP's goal.

---

[1] Video of learned policies: `https://youtu.be/MGvvPmm6JQg`
[2] Code: `https://github.com/deep-skill-chaining/deep-skill-chaining`

One way to learn a policy in an MDP is to first learn an action-value function. The action-value function $Q^\pi(s_t, a_t)$ is defined as the expected sum of discounted future rewards if the agent takes action $a_t$ from $s_t$ and then follows policy $\pi$ thereafter: $Q^\pi(s_t, a_t) = \mathbb{E}_\pi[r_t + \gamma \max_{a_{t+1}} Q^\pi(s_{t+1}, a_{t+1})]$.

Q-learning (Watkins & Dayan, 1992) is a commonly used off-policy algorithm that uses the action-value function for control through a greedy policy $\pi(s_t) = \arg\max_{a_t} Q(s_t, a_t)$. Inspired by recent success in scaling Q-learning to high-dimensional spaces (Mnih et al., 2015; Van Hasselt et al., 2016; Lillicrap et al., 2015; Tesauro, 1994), we learn the action-value function $Q_\phi^\pi(s_t, a_t)$ using non-linear function approximators parameterized by $\phi$, by minimizing the loss $L(\phi) = \mathbb{E}_\pi[(Q_\phi(s_t, a_t) - y_t)^2]$ where the Q-learning target $y_t$ is given by the following equation (Van Hasselt et al., 2016):

$$y_t = r_t + \gamma Q_{\phi'}(s_{t+1}, \arg\max_{a_{t+1}} Q_\phi(s_{t+1}, a_{t+1})). \tag{1}$$

Deep Q-Learning (DQN) (Mnih et al., 2015) casts minimizing $L(\phi)$ as a standard regression problem by using target networks (parameterized by $\phi'$) and experience replay (Lin, 1993).

## 2.1 THE OPTIONS FRAMEWORK

The options framework (Sutton et al., 1999) models skills as *options*. An option $o$ consists of three components: (a) its initiation condition, $\mathcal{I}_o(s)$, which determines whether $o$ can be executed in state $s$, (b) its termination condition, $\beta_o(s)$, which determines whether option execution must terminate in state $s$ and (c) its closed-loop control policy, $\pi_o(s)$, which maps state $s$ to a low level action $a \in A$. Augmenting the set of available actions with options results in a Semi-Markov Decision Process (SMDP) (Sutton et al., 1999) where the next state depends on the current state, action *and* time.

## 2.2 SKILL DISCOVERY ALGORITHMS

Skill discovery has been studied extensively in small discrete domains (McGovern & Sutton, 1998; Şimşek & Barto, 2004; Şimşek et al., 2005; Bakker & Schmidhuber, 2004; Schmidhuber, 1991; Pickett & Barto, 2002; Dietterich, 2000). Recently however, there has been a significant body of work aimed at discovering skills in continuous spaces.

**Option-critic methods**: Option-Critic (Bacon et al., 2017) uses an end-to-end gradient based algorithm to learn options in high-dimensional continuous spaces. Option-Critic was a substantial step forward in skill discovery and led to a family of related methods (Klissarov et al., 2017; Tiwari & Thomas, 2019; Riemer et al., 2018; Liu et al., 2017; Jain et al., 2018). Proximal Policy Option Critic (PPOC) (Klissarov et al., 2017) extends Option-Critic to continuous action spaces and is the version of Option-Critic that we compare against in this paper. Our method bypasses two fundamental shortcomings of the Option-Critic framework: (a) unlike Option-Critic, we explicitly learn initiation sets of options and thus do not assume that all options are executable from everywhere, and (b) we do not treat the number of skills required to solve a task as a fixed and costly hyperparameter. Instead, our algorithm flexibly discovers as many skills as it needs to solve the given problem.

**Feudal methods**: An alternative to the options framework is Feudal RL (Dayan & Hinton, 1993), which creates a hierarchy in which *managers* learn to assign subgoals to *workers*; workers take a subgoal state as input and learn to reach it. Feudal Networks (FuN) (Vezhnevets et al., 2017) used neural networks to scale the Feudal-RL framework to high-dimensional continuous spaces; it was extended and outperformed by HIRO (Nachum et al., 2018) in a series of control tasks in the MuJoCo simulator. More recently, Hierarchical Actor-Critic (HAC) (Levy et al., 2019) outperformed HIRO in a similar suite of continuous control problems. While HIRO relies on having a dense "distance-to-goal" based reward function to train both levels of their feudal hierarchy, HAC's use of Hindsight Experience Replay (HER) (Andrychowicz et al., 2017) allows it to work in the more general sparse-reward setting. Given its strong performance in continuous control problems and its ability to learn effectively in sparse-reward settings, we compare against HAC as a representative feudal method.

**Learning backward from the goal**: The idea of sequencing locally applicable controllers is well established in robotics and control theory in the form of pre-image backchaining (Kaelbling & Lozano-Pérez, 2017) and LQR-Trees (Tedrake, 2009). Such methods either require individually engineered control loops or a model of the system dynamics. Our work fits in the model-free RL setting and

thus requires neither. More recently, reverse curriculum learning (Florensa et al., 2017) also learns backward from the goal. However, they define a curriculum of start states to learn a single policy, rather than learning skills. Relay Networks (Kumar et al., 2018) segment the value function backward from the goal using a thresholding scheme, which makes their method reliant on the accurate estimation of the value function. By contrast, our algorithm is agnostic to errors in value estimation, which are unavoidable when using function approximation in high-dimensional spaces.

**Planning with learned skills**: Options have been shown to empirically speed up planning in several domains (Silver & Ciosek, 2012; Jinnai et al., 2019; James et al., 2018; Francis & Ram, 1993; Konidaris, 2016; Sharma et al., 2019). However, Konidaris et al. (2018) show that for resulting plans to be *provably* feasible, skills must be executable sequentially. While they assume that such skills are given, we show that they can be autonomously discovered in high-dimensional spaces.

## 3 DEEP SKILL CHAINING

Deep skill chaining (DSC) is based on the intuition that it is easier to solve a long-horizon task from states in the local neighborhood of the goal. This intuition informs the first step of the algorithm: create an option that initiates near the goal and reliably takes the agent to the goal. Once such an option is learned, we create another option whose goal is to take the agent to a state from which it can successfully execute the first option. Skills are chained backward in this fashion until the start state of the MDP lies inside the initiation set of some option. The inductive bias of creating sequentially executable skills guarantees that as long as the agent successfully executes each skill in its chain, it will solve the original task. More formally, skill chaining amounts to learning options such that the termination condition $\beta_{o_i}(s_t)$ of an option $o_i$ is the initiation condition $\mathcal{I}_{o_{i-1}}(s_t)$ of the option that precedes it in its chain.

Our algorithm proceeds as follows: at time $t$, the policy over options $\pi_{\mathcal{O}} : s_t \in S \rightarrow o \in \mathcal{O}$ determines which option to execute (Section 3.2). Control is then handed over to the selected option $o_i$'s internal policy $\pi_{o_i} : s \in S \rightarrow a_t \in \mathbb{R}^{|A|}$. $\pi_{o_i}$ outputs joint torques until it either reaches its goal ($\beta_{o_i} := \mathcal{I}_{o_{i-1}}$) or times out at its predetermined budget $T$ (Section 3.1). At this point, $\pi_{\mathcal{O}}$ chooses another option to execute. If at any point the agent reaches the goal state of the MDP or the initiation condition of a previously learned option, it creates a new option to target such a salient event. The machinery for learning the initiation condition of this new option is described in Section 3.3. We now detail the components of our architecture and how they are learned. Readers may also refer to Figures 4 & 7 and the pseudo-code in Appendix A.5 to gain greater intuition about our algorithm.

### 3.1 INTRA-OPTION POLICY

Each option $o$ maintains its own policy $\pi_o : s \rightarrow a_t \in \mathbb{R}^{|A|}$, which is parameterized by its own neural networks $\theta_o$. To train $\pi_o(s; \theta_o)$, we must define $o$'s internal reward function. In sparse reward problems, $o$ is given a subgoal reward when it triggers $\beta_o$; otherwise it is given a step penalty. In the dense reward setting, we can compute the distance to the parent option's initiation set classifier and use that to define $o$'s internal reward function. We can now treat learning the intra-option policy ($\pi_o$) as a standard RL problem and use an off-the-shelf algorithm to learn this policy. Since in this work we solve tasks with continuous action spaces, we use Deep Deterministic Policy Gradient (DDPG) (Lillicrap et al., 2015) to learn option policies over real-valued actions.

### 3.2 POLICY OVER OPTIONS

Initially, the policy over options ($\pi_{\mathcal{O}}$) only possesses one option that operates over a single time step ($T = 1$). We call this option the *global option* ($o_G$) since its initiation condition is true everywhere in the state space and its termination condition is true only at the goal state of the MDP (i.e, $\mathcal{I}_{o_G}(s) = 1 \forall s$ and $\beta_{o_G} = \mathbb{1}_g$). Using $o_G$, $\pi_{\mathcal{O}}$ can select primitive actions. At first the agent continually calls upon $o_G$, which uses its internal option policy $\pi_{o_G}$ to output exactly one primitive action. Once $o_G$ triggers the MDP's goal state $N$ times, DSC creates its first temporally extended option, the *goal option* ($o_g$), whose termination condition is also set to be the goal state of the MDP, i.e, $\beta_{o_g} = \mathbb{1}_g$.

As the agent discovers new skills, it adds them to its option repertoire and relies on $\pi_{\mathcal{O}}$ to determine which option (including $o_G$) it must execute at each state. Unlike $o_G$, learned options will be

temporally extended, i.e, they will operate over $T > 1$ time steps. If in state $s_t$ the agent chooses to execute option $o_i$, then $o_i$ will execute its own closed-loop control policy (for $\tau$ steps) until its termination condition is met ($\tau < T$) or it has timed out at $\tau = T$ time steps. At this point, control is handed back to $\pi_{\mathcal{O}}$, which must now choose a new option at state $s_{t+\tau}$.

**Option selection**: To select an option in state $s_t$, $\pi_{\mathcal{O}}$ first constructs a set of admissible options given by Equation 2. $\pi_{\mathcal{O}}$ then chooses the admissible option that maximizes its option-value function, as shown in Equation 3. Since the agent must choose from a discrete set of options at any time, we learn its option-value function using Deep Q-learning (DQN) (Mnih et al., 2015).

$$\mathcal{O}'(s_t) = \{o_i | \mathcal{I}_{o_i}(s_t) = 1 \cap \beta_{o_i}(s_t) = 0, \forall o_i \in \mathcal{O}\} \quad (2)$$

$$o_t = \arg\max_{o_i \epsilon \mathcal{O}'(s_t)} Q_\phi(s_t, o_i). \quad (3)$$

**Learning the option-value function**: Given an SMDP transition $(s_t, o_t, r_{t:t+\tau}, s_{t+\tau})$, we update the value of taking option $o_t$ in state $s_t$ according to SMDP Q-learning update (Bradtke & Duff, 1995). Since the agent learns Q-values for different state-option pairs, it may choose to ignore learned options in favor of primitive actions in certain parts of the state-space (in the interest of maximizing its expected future sum of discounted rewards). The Q-value target for learning the weights $\phi$ of the DQN is given by:

$$y_t = \sum_{t'=t}^{\tau} \gamma^{t'-t} r_{t'} + \gamma^{\tau-t} Q_{\phi'}(s_{t+\tau}, \arg\max_{o'\epsilon\mathcal{O}'(s_{t+\tau})} Q_\phi(s_{t+\tau}, o')). \quad (4)$$

**Adding new options to the policy over options**: Equations 2, 3 and 4 show how we can learn the option-value function and use it for selecting options. However, we must still incrementally add *new* skills to the network during the agent's lifetime. After the agent has learned a new option $o$'s initiation set classifier $\mathcal{I}_o$ (we will discuss how this happens in Section 3.3), it performs the following steps before it can add $o$ to its option repertoire:

- To initialize $o$'s internal policy $\pi_o$, the parameters of its DDPG ($\theta_o$) are set to the parameters of the global agent's DDPG ($\theta_{o_G}$). Subsequently, their neural networks are trained independently. This provides a good starting point for optimizing $\pi_o$, while allowing it to learn sub-problem specific abstractions.

- To begin predicting Q-values for $o$, we add a new output node to final layer of the DQN parameterizing $\pi_{\mathcal{O}}$.

- We must assign appropriate initial values to $Q_\phi(s, o)$. We follow Konidaris & Barto (2009b) and collect all the transitions that triggered $\beta_o$ and use the max over these Q-values to optimistically initialize the new output node of our DQN.[3] This is done by setting the bias of this new node, which ensures that the Q-value predictions corresponding to the other options remain unchanged.

### 3.3 INITIATION SET CLASSIFIER

Central to the idea of learning skills is the ability to learn the set of states from which they can be executed. First, we must learn the initiation set classifier for $o_g$, the option used to trigger the MDP's goal state. While acting in the environment, the agent's global DDPG will trigger the goal state $N$ times (also referred to as the gestation period of the option by Konidaris & Barto (2009b) and Niekum & Barto (2011)). We collect these $N$ successful trajectories, segment the last $K$ states from each trajectory and learn a one-class classifier around the segmented states. Once initialized, it may be necessary to refine the option's initiation set based on its policy. We do so by executing the option and collecting data to train a two-class classifier. States from which option execution was successful are labeled as positive examples. States from which option execution timed out are labeled as negative examples. We continue this process of refining the option's initiation set classifier for a fixed number of episodes, which we call the initiation period of the option.

---

[3]Using the mean Q-value is equivalent to performing Monte Carlo rollouts. Instead, we follow the principle of *optimism under uncertainty* (Brafman & Tennenholtz, 2002) to select the max over the Q-values.

At the end of the initiation period, we fix the option's initiation set classifier and add it to the list of salient events in the MDP. We then construct a new option whose termination condition is the initiation classifier of the option we just learned. We continue adding to our chain of options in this fashion until a learned initiation set classifier contains the start state of the MDP.

## 3.4 Generalizing to Skill Trees

Our discussion so far has been focused on learning skill chains that extend from the goal to the start state of the MDP. However, such a chain is not sufficient if the agent has multiple start states or if we want the agent to learn multiple ways of solving the same problem. To permit such behavior, our algorithm can be used to learn skills that organize more generally in the form of trees (Konidaris & Barto, 2009b; Konidaris et al., 2012). This generalization requires some additional care while learning initiation set classifiers, the details of which can be found in Section A.1 of the Appendix. To demonstrate our ability to construct such skill trees (and their usefulness), we consider a maze navigation task, E-Maze, with distinct start states in Section 4.

## 3.5 Optimality of Discovered Solutions

Each option $o$'s internal policy $\pi_o$ is is given a subgoal reward only when it triggers its termination condition $\beta_o$. As a result, $\pi_o$ is trained to find the optimal trajectory for entering its own goal region. Naively executing learned skills would thus yield a *recursively optimal* solution to the MDP (Barto & Mahadevan, 2003). However, since the policy over options $\pi_\mathcal{O}$ does not see subgoal rewards and is trained using extrinsic rewards only, it can combine learned skills and primitive actions to discover a *flat optimal* solution $\pi^*$ to the MDP (Barto & Mahadevan, 2003). Indeed, our algorithm allows $\pi_\mathcal{O}$ to employ discovered skills to quickly and reliably find feasible paths to the goal, which over time can be refined into optimal solutions. It is worth noting that our ability to recover $\pi^*$ in the limit is in contrast to feudal methods such as HAC (Levy et al., 2019) in which higher levels of the hierarchy are rewarded for choosing feasible subgoals, not optimal ones.

To summarize, our algorithm proceeds as follows: (1) Collect trajectories that trigger new option $o_k$'s termination condition $\beta_{o_k}$. (2) Train $o_k$'s option policy $\pi_{o_k}$. (3) Learn $o_k$'s initiation set classifier $\mathcal{I}_{o_k}$. (4) Add $o_k$ to the agent's option repertoire. (5) Create a new option $o_{k+1}$ such that $\beta_{o_{k+1}} = \mathcal{I}_{o_k}$. (6) Train policy over options $\pi_\mathcal{O}$. Steps 1, 3, 4 and 5 continue until the MDP's start state is inside some option's initiation set. Continue steps 2 and 6 indefinitely.

## 4 Experiments

We test our algorithm in five tasks that exhibit a strong hierarchical structure: (1) Point-Maze (Duan et al., 2016), (2) Four Rooms with Lock and Key, (3) Reacher (Brockman et al., 2016), (4) Point E-Maze and (5) Ant-Maze (Duan et al., 2016; Brockman et al., 2016). Since tasks 1, 3 and 5 appear frequently in the literature, details of their setup can be found in Appendix A.3.

**Four Rooms with Lock and Key**: In this task, a point agent (Duan et al., 2016) is placed in the Four Rooms environment (Sutton et al., 1999). It must pick up the key (blue sphere in the top-right room in Figure 1(c), row 2) and *then* navigate to the lock (red sphere in the top-left room). The agent's state space consists of its position, orientation, linear velocity, rotational velocity and a `has_key` indicator variable. If it reaches the lock with the key in its possession, its episode terminates with a sparse reward of 0; otherwise it gets a step penalty of $-1$. If we wish to autonomously discover the importance of the key, (i.e, without any corresponding extrinsic rewards) a distance-based dense reward such as that used in related work (Nachum et al., 2018) would be infeasible.

**Point E-Maze**: This task extends the benchmark U-shaped Point-Maze task (Duan et al., 2016) so that the agent has two possible start locations - on the top and bottom rungs of the E-shaped maze respectively. We include this task to demonstrate our algorithm's ability to construct skill trees.

## 4.1 Comparative Analyses

We compared the performance of our algorithm to DDPG, Option-Critic and Hierarchical Actor-Critic (HAC), in the conditions most similar to those in which they were originally evaluated. For

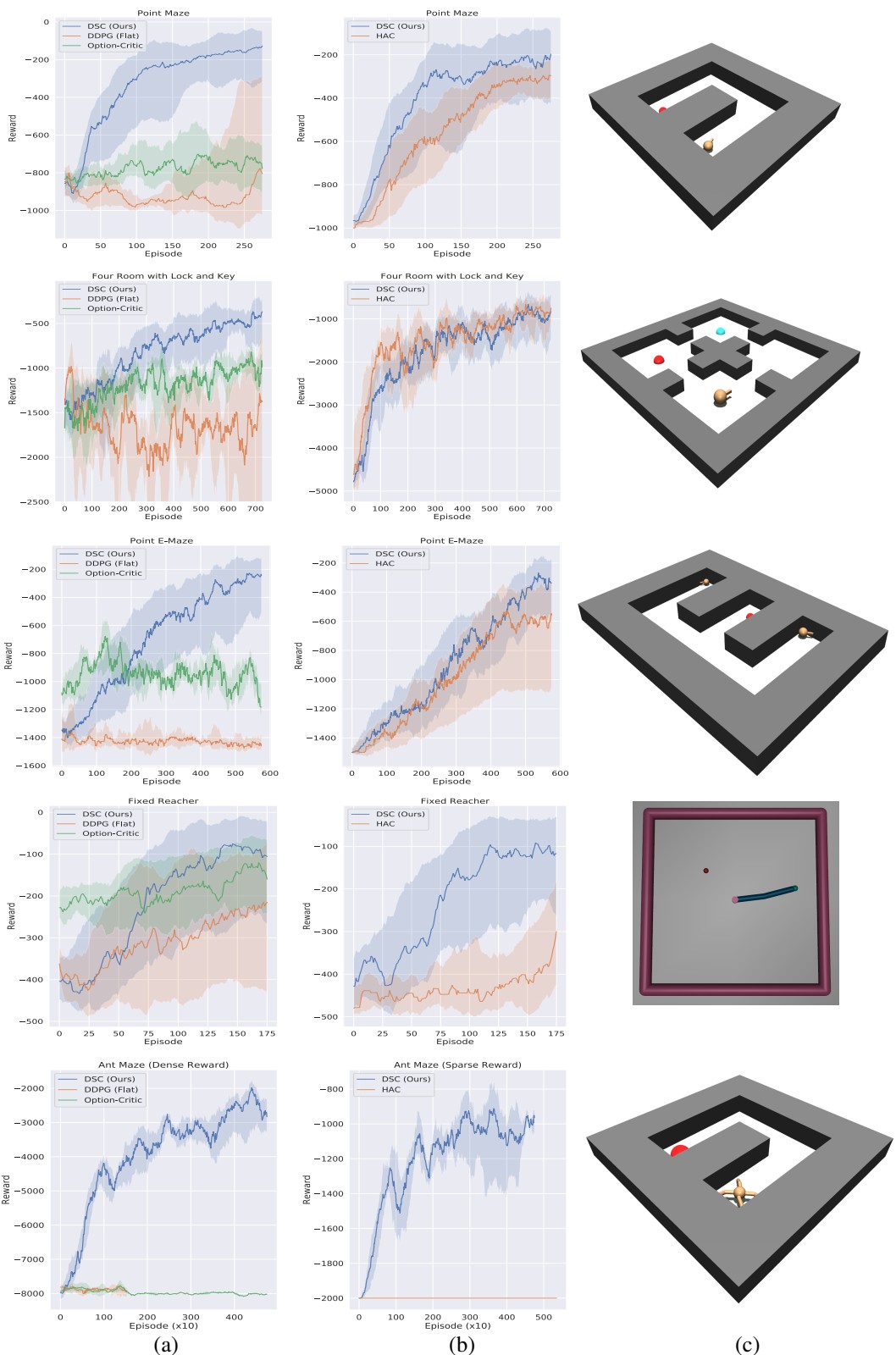

Figure 1: (a) Learning curves comparing deep skill chaining (DSC), a flat agent (DDPG) and Option-Critic. (b) Comparison with Hierarchical Actor Critic (HAC). (c) the continuous control tasks corresponding to the learning curves in (a) and (b). Solid lines represent median reward per episode, with error bands denoting one standard deviation. Our algorithm remains the same between (a) and (b). All curves are averaged over 20 runs, except for Ant Maze which was averaged over 5 runs.

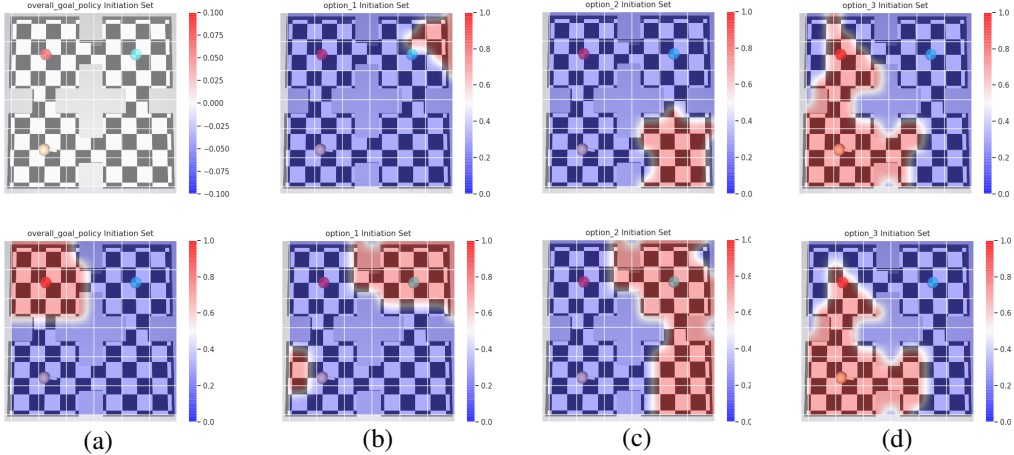

Figure 2: Initiation sets of options learned in the Lock and Key task. Blue sphere in top-right room represents the key, red sphere in top-left room represents the lock. Red regions represent states inside the initiation classifier of learned skills, whereas blue/gray regions represent states outside of it. Each column represents an option - the top row corresponding to the initiation set when `has_key` is false and the bottom row corresponding to the initiation set when `has_key` is true.

instance, in the Ant-Maze task we compare against Option-Critic under a dense-reward formulation of the problem while comparing to HAC under a sparse-reward version of the same task. As a result, we show the learning curves comparing against them on different plots (columns (a) and (b) in Figure 1 respectively) to emphasize the difference between the algorithms, the settings in which they are applicable, and the way they are evaluated.

**Comparison with DDPG and Option-Critic**: Figure 1(a) shows the results of comparing our proposed algorithm (DSC) with a flat RL agent (DDPG) and the version of Option-Critic designed for continuous action spaces (PPOC).[4] Deep skill chaining comfortably outperforms both baselines. Both DSC and DDPG use the same exploration strategy in which $a_t = \pi_\theta(s_t) + \eta_t$ where $\eta_t \sim N(0, \epsilon_t)$. Option-Critic, on the other hand, learns a stochastic policy $\pi_\theta(a_t|s_t)$ and thus has baked-in exploration (Sutton & Barto, 2018, Ch. 13), precluding the need for additive noise during action selection. We hypothesize that this difference in exploration strategies is the reason Option-Critic initially performs better than both DDPG and DSC in the Reacher and Point E-Maze tasks.

**Comparison with Hierarchical Actor-Critic**: We compare our algorithm to Hierarchical Actor-Critic (HAC) (Levy et al., 2019), which has recently outperformed other hierarchical reinforcement learning methods (Nachum et al., 2018; Vezhnevets et al., 2017) on a wide variety of tasks. [5] A noteworthy property of the HAC agent is that it may prematurely terminate its training episodes to prevent flooding its replay buffer with uninformative transitions. The length of each training episode in DSC however, is fixed and determined by the test environment. Unless the agent reaches the goal state, its episode lasts for the entirety of its episodic budget (e.g, this would be 1000 timesteps in the Point-Maze environment). Thus, to compare the two algorithms, we perform periodic test rollouts wherein all networks are frozen and both algorithms have the same time budget to solve the given task. Furthermore, since both DSC and HAC learn deterministic policies, we set $\epsilon_t = 0$ during these test rollouts. When comparing to HAC, we perform 1 test rollout after each training episode in all tasks except for Ant-Maze, where we average performance over 5 test rollouts every 10 episodes.

Figure 1(b) shows that DSC outperforms HAC in all environments except for Four Rooms with a Lock and Key, where their performance is similar, even though DSC does not use Hindsight Experience Replay (Andrychowicz et al., 2017) to deal with the sparse reward nature of this task.

## 4.2 INTERPRETING LEARNED SKILLS

Figure 2 visualizes the initiation set classifiers of options discovered by DSC in Four Rooms with a Lock and Key. Despite not getting any extrinsic reward for picking up the key, DSC discovers

---

[4]PPOC author's implementation: https://github.com/mklissa/PPOC

[5]HAC author's implementation: https://github.com/andrew-j-levy/Hierarchical-Actor-Critc-HAC-

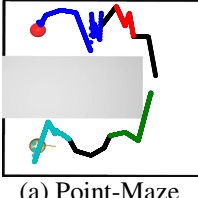 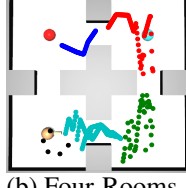 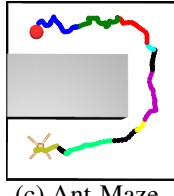 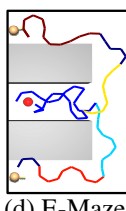

(a) Point-Maze     (b) Four-Rooms     (c) Ant-Maze     (d) E-Maze

Figure 3: Solution trajectories found by deep skill chaining. Sub-figure (d) shows two trajectories corresponding to the two possible initial locations in this task. Black points denote states in which $\pi_{\mathcal{O}}$ chose primitive actions, other colors denote temporally extended option executions.

the following skill chain: the options shown in Figure 2 columns (c) and (d) bring the agent to the room with the key. The option shown in column (b) then picks up the key (top row) and then takes the agent to the room with the lock (bottom row). Finally, the option in column (a) solves the overall problem by navigating to the lock with the key. Similar visualizations of learned initiation set classifiers in the E-Maze task can be found in the Figure 6 in the Appendix.

Figure 3 shows that DSC is able to learn options that induce simple, efficient policies along different segments of the state-space. Furthermore, it illustrates that in some states, the policy over options prefers primitive actions (shown in black) over learned skills. This suggests that DSC is robust to situations in which it constructs poor options or is unable to learn a good option policy in certain portions of the state-space. In particular, Figure 3 (d) shows how DSC constructs a skill tree to solve a problem with two distinct start states. It learns a common option near the goal (shown in blue), which then branches off into two different chains leading to its two different start states respectively.

## 5 DISCUSSION AND CONCLUSION

Deep skill chaining breaks complex long-horizon problems into a series of sub-problems and learns policies that solve those sub-problems. By doing so, it provides a significant performance boost when compared to a flat learning agent in all of the tasks considered in Section 4.

We show superior performance when compared to Option-Critic, the leading framework for option discovery in continuous domains. A significant drawback of Option-Critic is that it assumes that all options are executable from everywhere in the state-space. By contrast, deep skill chaining explicitly learns initiation set classifiers. As a result, learned skills specialize in different regions of the state-space and do not have to bear the burden of learning representations for states that lie far outside of their initiation region. Furthermore, each option in the Option-Critic architecture leverages the same state-abstraction to learn option-specific value functions and policies, while deep skill chaining permits each skill to construct its own skill-specific state-abstraction (Konidaris & Barto, 2009a). An advantage of using Option-Critic over DSC is that it is not confined to goal-oriented tasks and can work in tasks which require continually maximizing non-sparse rewards.

Section 4 also shows that deep skill chaining outperforms HAC in four out of five domains, while achieving comparable performance in one. We note that even though HAC was designed to work in the multi-goal setting, we test it here in the more constrained single-goal setting. Consequently, we argue that in problems which permit a stationary set of target events (like the ones considered here), deep skill chaining provides a favorable alternative to HAC. Furthermore, HAC depends on Hindsight Experience Replay (HER) to train the different layers of their hierarchy. Deep skill chaining shows the benefits of using hierarchies even in the absence of such data augmentation techniques but including them should yield additional performance benefits in sparse-reward tasks.

A drawback of deep skill chaining is that, because it builds skills backward from the goal, its performance in large state-spaces is dependent on a good exploration algorithm. We used the naive exploration strategy of adding Gaussian noise to chosen actions (Lillicrap et al., 2015; Fujimoto et al., 2018) since the exploration question is orthogonal to the ideas presented here. The lack of a sophisticated exploration algorithm also explains the higher variance in performance in the Point-Maze task in Figure 1. Combining effective exploration (Machado et al., 2018; Jinnai et al., 2020) with DSC's high reliability of triggering target events is a promising avenue for future work.

We presented a new skill discovery algorithm that can solve high-dimensional goal-oriented tasks far more reliably than flat RL agents and other popular hierarchical methods. To our knowledge,

DSC is the first deep option discovery algorithm that does not treat the number of options as a fixed and costly hyperparameter. Furthermore, where other deep option discovery techniques have struggled to show consistent improvements over baseline flat agents in the single task setting (Zhang & Whiteson, 2019; Smith et al., 2018; Harb et al., 2018; Klissarov et al., 2017), we unequivocally show the necessity for hierarchies for solving challenging problems.

## 6 ACKNOWLEDGEMENTS

We thank Andrew Levy, Nakul Gopalan, Sam Lobel, Theresa Barton and other members of the Brown bigAI group for their inputs. This research was supported in part by DARPA under agreement number W911NF1820268, AFOSR Young Investigator Grant agreement number FA9550-17-1-0124 and the ONR under the PERISCOPE MURI Contract N00014-17-1-2699. The U.S. Government is authorized to reproduce and distribute reprints for Governmental purposes notwithstanding any copyright notation thereon. The content is solely the responsibility of the authors and does not necessarily represent the official views of DARPA, the ONR, or the AFOSR.

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

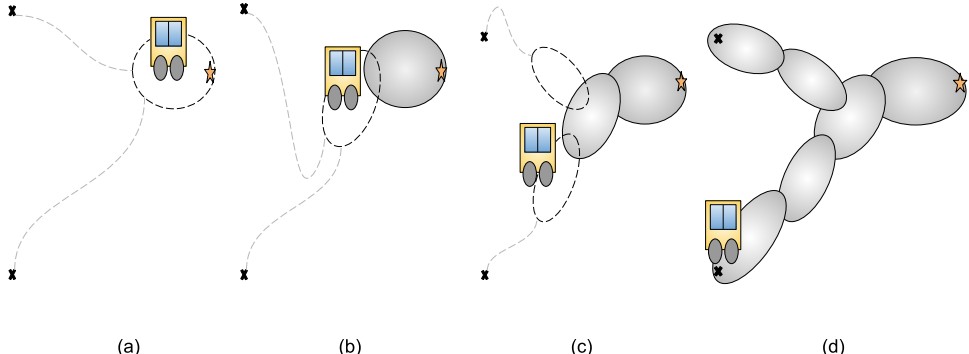

Figure 4: An illustration of the deep skill chaining algorithm. $\star$ represents the goal state, $\times$ represents the two start states. (a) Before the agent has discovered its first skill/option, it acts according to its global DDPG policy. Having encountered the goal state $N$ times, the agent creates an option to trigger the goal from its local neighborhood. (b) Now, when the agent enters the initiation set of the first option, it begins to learn another option to trigger the first option. (c) Because the agent has two different start states, it learns two qualitatively different options to trigger the option learned in (b). (d) Finally, the agent has learned a skill tree which it can follow to consistently reach the goal.

## A  APPENDIX

### A.1  CREATING SKILL TREES

In Section 3.4, we introduced the idea of generalizing skill chains to skill trees to incorporate qualitatively different solution trajectories. In this section, we provide some of the implementation details required to learn initiation set classifiers that organize in the form of trees.

When creating skill chains, the goal of each option is to trigger the initiation condition of the option that precedes it in its chain (i.e, its parent option). When creating a skill tree of branching factor $B$, we allow at most $B$ options to target each salient event in the MDP (i.e, the goal state and the initiation set classifiers of preexisting options). To further control the branching factor of the skill tree, we impose two more conditions on option creation:

1. Consider an option $o_1$ which already has one child option $o_2$ targeting it. Now suppose that we want to learn another option $o_3$ that also targets $o_1$. We only consider state $s_t$ to be a positive example for training $\mathcal{I}_{o_3}$ if $\mathcal{I}_{o_2}(s_t) = 0$.

2. To prevent significant overlap between options that target the same event, we treat the positive examples used to train the initiation set classifier of one as negative training examples of all its sibling options. This allows for multiple options that trigger the same target event, while encouraging them to specialize in different parts of the state-space.

In the Point E-Maze task considered in Section 4, we learn a skill tree with $B = 2$.

### A.2  INTRA-OPTION Q-LEARNING

In principle, the methodology outlined in Section 3.2 is sufficient to learn an effective policy over options $\pi_{\mathcal{O}}$. However, when $\mathcal{O}$ is a set of Markov options (Sutton et al., 1999), which is the setting considered in this paper, we can use intra-option Q-learning (Sutton et al., 1998) to improve the sample efficiency associated with learning $\pi_{\mathcal{O}}$.

More specifically, given a transition $(s_t, o, r_{t:t+\tau}, s_{t+\tau})$, SMDP Q-learning treats option $o$ as a black box and uses Equation 4 to determine the Q-value target $y_t$ for updating $\pi_{\mathcal{O}}$. Intra-option Q-learning leverages the fact that option $o$ is Markov to point out that all the transitions experienced during the execution of $o$ are also valid experiences for training $\pi_{\mathcal{O}}$. As long as a state $s_{t+i}, \forall i \in [0, \tau]$ is inside the initiation set of the option $o$, we can pretend that option execution really began in state $s_{t+i}$ and add the transition $(s_{t+i}, o, r_{t+i:t+\tau}, s_{t+\tau})$ to the $\pi_{\mathcal{O}}$'s replay buffer.

Furthermore, intra-option Q-learning also provides a way to improve the sample efficiency associated with learning option policies $\pi_o, \forall o \in \mathcal{O}$. This can be done by making off-policy updates to each option's internal policy. In other words, regardless of which option is actually executed in the MDP, as long as a state experienced during execution is inside the initiation set of some other option, we can add the associated experience tuple to that (un-executed) option's replay buffer. Note that this is possible because we use an off-policy learning algorithm (DDPG) to learn intra-option policies.

## A.3 TEST ENVIRONMENTS

A description of the Four Rooms and the Point E-Maze tasks was provided in Section 4. Here we describe the remaining tasks considered in this paper:

**Point Maze**: In this task, the same point agent as in the four rooms task must navigate around a U-shaped maze to reach its goal. The agent receives a reward of $-1$ for every step it lives, and a sparse terminating reward of $0$ when it reaches its goal location. This is an interesting task for hierarchical agents because in order to reach the goal, the agent must first move away from it. It is clear that a dense distance-based reward formulation of this problem would only serve to deceive non-hierarchical agents such as DDPG.

**Ant Maze**: The ant (Duan et al., 2016) is a challenging agent to control due to its non-linear and highly unstable dynamics. In this task, the ant must now navigate around the same U-shaped maze as in the Point Maze task. Getting the ant to cover significant distances along the $x, y$ plane without falling over, is a benchmark control task itself (Brockman et al., 2016). As a result, constructing options backward from the goal could require prohibitively large training episodes or the use of a sophisticated exploration algorithms (Burda et al., 2019; Bellemare et al., 2016; Tang et al., 2017). To avoid conflating our results with the orthogonal investigation of effective exploration in RL, we follow the experimental design of other state-of-the-art hierarchical reinforcement learning algorithms (Levy et al., 2019; Nachum et al., 2018) and sample the initial state of the ant uniformly across the maze for the first 30 episodes. For fair comparison, all baseline algorithms use this exploration strategy.

**Fixed Reacher**: We use the Reacher task (Brockman et al., 2016) with two modifications. First, rather than randomly sampling a new goal at the start of each episode, we fix the target across all episodes. We do this because if the goal moves, following a learned skill chain will no longer solve the MDP. Note that the same modification was made in the DDPG paper (Lillicrap et al., 2015). Second, to increase the difficulty of the resulting task, we use a sparse reward function rather than the dense distance-based one used in the original formulation.

| Task | Number of steps per episode |
|------|----------------------------:|
| Point-Maze | 1000 |
| Four Rooms with Lock and Key | 5000 |
| Point E-Maze | 1500 |
| Reacher | 500 |
| Ant-Maze | 2000 |

Table 1: Maximum number of time steps per episode in each of the experimental domains

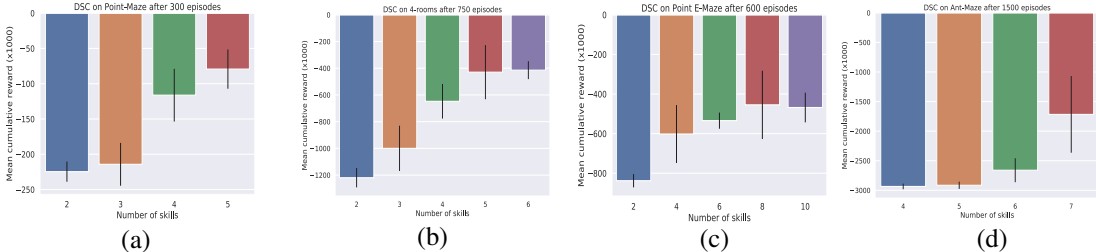

Figure 5: Analysis of performance (as measured by mean cumulative reward) of DSC agent as it is allowed to learn more skills in (a) Point-Maze, (b) Four Rooms with Lock and Key, (c) E-Maze and (d) Ant-Maze. Note that in general, DSC discovers as many skills as it needs to solve the given problem. For this experiment alone, we restrict the number of skills that the DSC agent can learn. All experiments averaged over 5 runs. Error bars denote 1 standard deviation. Higher is better.

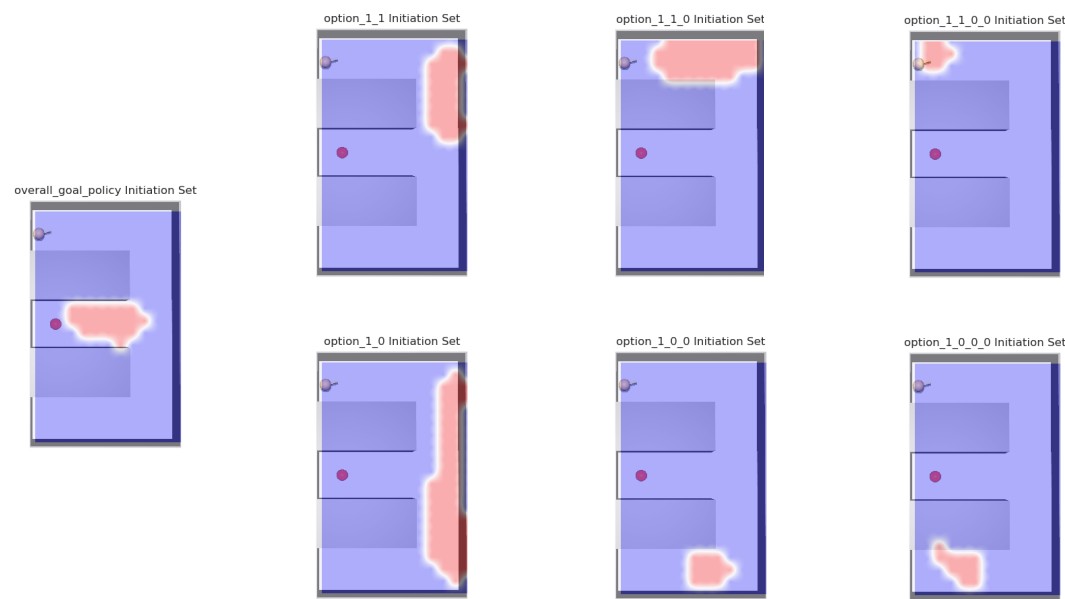

Figure 6: Initiation set classifiers learned in the Point E-Maze domain. Discovered skills organize in the form of a tree with a branching factor of 2. The option on the extreme left initiates in the proximity of the goal. Options learned after the goal option branch off into two separate skill chains. The chain on top extends backward to the start state in the top rung of the E-Maze. The chain shown in the bottom row extends backward to the start state in the bottom rung of the E-Maze.

## A.4 ABLATION STUDY

### A.4.1 PERFORMANCE AS A FUNCTION OF NUMBER OF SKILLS

Deep skill chaining generally discovers and learns as many skills as it needs to solve a given problem. In this experiment however, we restrict the number of skills DSC can learn to examine its impact on overall agent performance (as measured by cumulative reward during training). Figure 5 shows that the performance of the agent increases monotonically (with diminishing marginal improvements) as it is allowed to learn more skills.

### A.4.2 NUMBER OF SKILLS OVER TIME

Figures 7 (a) and 7 (b) illustrate how deep skill chaining incrementally discovers options and adds it to the agent's option repertoire. Figure 7(c) shows how the number of skills empirically increases over time, plateaus and has low variance between runs. Since the agent has to learn the importance of the key in the Four Rooms task, learning initiation set classifiers takes longer than in the Point-Maze task.

### A.4.3 HYPERPARAMETER SENSITIVITY

In this section, we analyze DSC's sensitivity to some of the hyperparameters specific to the algorithm. In Figure 8, we show that even under a fairly large range of values for the buffer length $K$ and the gestation period $N$, DSC is able to retain its strong performance.

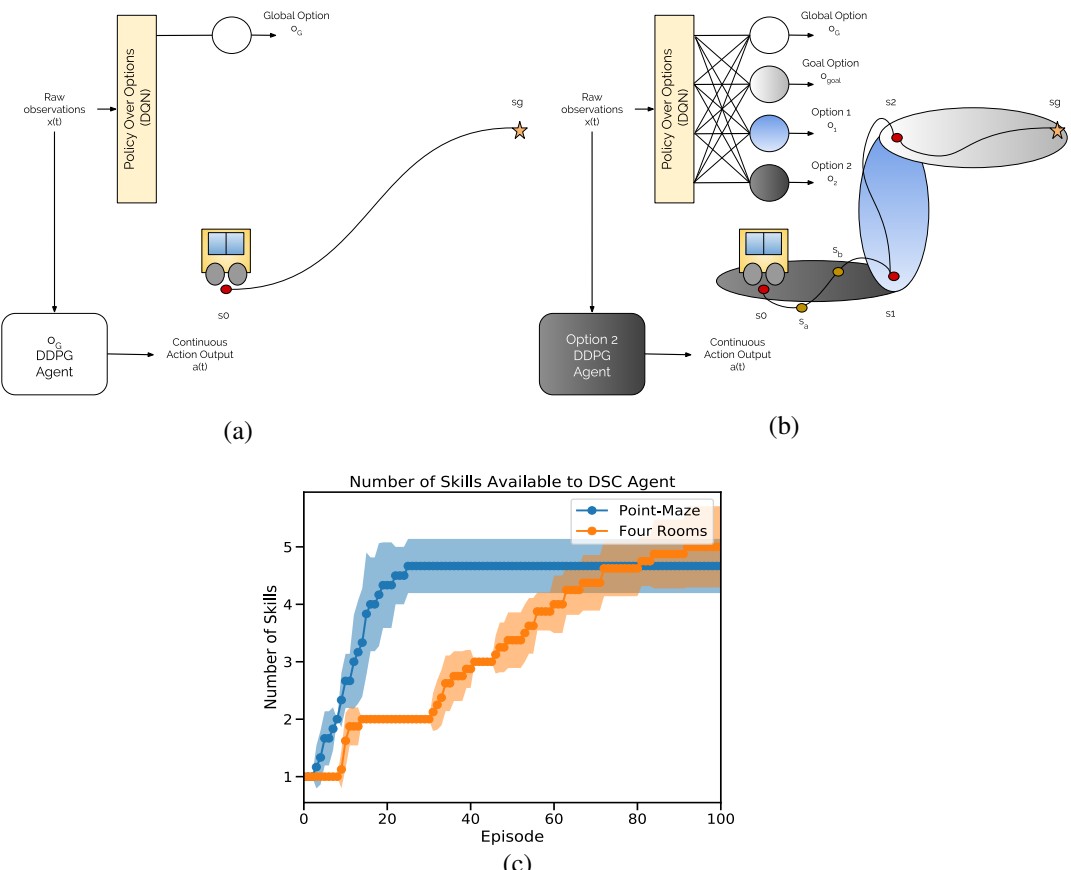

Figure 7: (a) Initially, the policy over options $\pi_{\mathcal{O}}$ can only choose the global option $o_G$ as a proxy for selecting primitive actions. (b) Over time, the agent learns temporally extended skills and adds output nodes to the final layer of the DQN parameterizing $\pi_{\mathcal{O}}$. This continues until the start state $s_0$ lies inside the initiation set of a learned option. (c) Empirical evaluation of how the number of skills in the agent's option repertoire changes over time in Point-Maze and Four-Rooms with a Lock and Key.

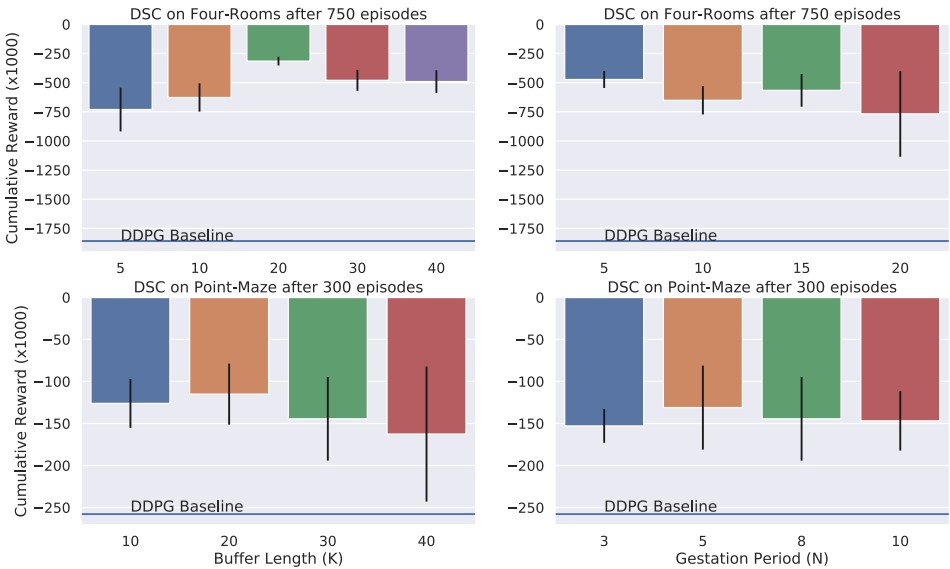

Figure 8: Variation in DSC performance (as measured by mean cumulative reward) as a function of two hyperparameters: (left) the buffer length $K$ and (right) the gestation period $N$ of the option. For a qualitative description of both hyperparameters, refer to Section 3.3. This experiment shows that DSC is fairly robust to most reasonable choices of these parameters. All experiments averaged over 5 runs. Error bars denote 1 standard deviation. Higher is better.

## A.5 ALGORITHM PSEUDO-CODE

---

**Algorithm 1:** Deep Skill Chaining

---

$s_0$ is the start state of the MDP
$\mathbb{1}_g(s) := 1$ if $s$ is a target state in the MDP, 0 otherwise
Given hyperparameter $T_0$, the time budget for discovered, temporally extended options
Global option: $o_G = (I_{o_G}, \pi_{o_G}, \beta_{o_G} = \mathbb{1}_g, T = 1)$
Goal option: $o_g = (I_{o_g}, \pi_{o_g}, \beta_{o_g} = \mathbb{1}_g, T = T_0)$
Agent's option repertoire: $\mathcal{O} = \{o_G\}$
Untrained Option: $o_U = o_g$ // option whose initiation classifier is yet unlearned
Policy over options: $\pi_{\mathcal{O}}: s_t \to o_t$
$s_t = s_0$
**while** *not $s_t$.is_terminal()* **do**

    **1. Pick new option and execute in environment**
    Choose $o_t$ according to $\pi_{\mathcal{O}}(s_t)$ using Equations 2 and 3
    $r_{t:\tau}, s_{t+\tau} = execute\_option(o_t)$
    $\pi_{\mathcal{O}}.update(s_t, o_t, r_{t:t+\tau}, s_{t+\tau})$ using Equation 4
    **2. Learn initiation set of new option**
    // Collect trajectories that trigger $o_U$'s termination region unless we have finished chaining
    **if** $\beta_{o_U}(s_{t+\tau})$ & $(s_0 \not\in I_{o_i} \forall o_i \in \mathcal{O})$ **then**
        $o_U.learn\_initiation\_classifier()$ using procedure described in Section 3.3
        **if** $o_U.initiation\_classifier\_is\_trained()$ **then**
            $\pi_{\mathcal{O}}.add(o_U)$ using procedure described in Section 3.2
            $\mathcal{O}.append(o_U)$
            $o_U = create\_child\_option(o_U)$
        **end**
    **end**
**end**
**Function** `create_child_option(o)`:
    """" Create a new option whose $\beta$ is the parent's $\mathcal{I}$. """"
    $o^*$ = Option() // Create a new option
    $\mathcal{I}_{o^*}$ = None
    $\beta_{o^*} = \mathcal{I}_o$
    **return** $o^*$
**Function** `execute_option(o_t)`:
    *""" Option control loop. """*
    $t_0 = t$
    *$T$ is the option's episodic time budget*
    *$\pi_{o_t}$ is the option's internal policy*
    **while** *not $\beta_{o_t}(s_t)$ & $t < T$* **do**
        $a_t = \pi_{o_t}(s_t; \theta_{o_t})$
        $r_t, s_{t+1} = env.step(a_t)$
        $s_t = s_{t+1}$
        $t = t + 1$
    **end**
    $\tau = t$ *// duration of option execution*
    **return** $r_{t_0:t_0+\tau}, s_{t_0+\tau}$

---

## A.6 MORE DETAILS ON IMPLEMENTING OPTION REWARD FUNCTIONS

Section 3.1 explains that to learn an option's intra-option policy, we must define its internal reward function. While most of our experiments are conducted in the sparse-reward setting, deep skill chaining can be used without much modification in dense reward tasks as well. All that remains is a clear description of how each option's internal reward function would be defined in such a setting.

Consider an option $o_i$ with parent option $o_{i-1}$ such that $\beta_{o_i} = \mathcal{I}_{o_{i-1}}$. In the dense reward setting, we use the negative distance from the state to the parent option's initiation classifier as the reward function. Since initiation classifiers are represented using parametric classifiers, computing the dis-

tance to the classifier's decision boundary is straightforward and can be done using most popular machine learning frameworks. For instance, when using scikit-learn (Pedregosa et al., 2011), this is implemented as follows:

$$R_o(s, a, s') = \begin{cases} 0, & \text{if } \beta_o(s') = 1 \\ -\mathcal{I}_{o_{i-1}}.\texttt{decision\_function}(s'), & \text{otherwise} \end{cases} \tag{5}$$

Where in Equation 5, `decision_function`$(x)$ returns the distance in feature space between point $x \in \mathbb{R}^N$ and the decision boundary learned by the classifier $\mathcal{I}_{o_{i-1}}$.

## A.7 LEARNING INITIATION SET CLASSIFIERS

To learn initiation set classifiers as described in Section 3.3, we used scikit-learn's One-Class SVM and Two-Class SVM packages (Pedregosa et al., 2011). Initiation set classifiers were learned on a subset of the state variables available in the domain. For instance, in the Lock and Key domain, the initiation set classifier was learned over the $x, y$ position and the `has_key` indicator variable. This is similar to other methods like HAC (Levy et al., 2019) which require the user to specify the dimensions of the state variable necessary to achieve the overall goal of the MDP. Incorporating the entire state variable to learn initiation set classifiers or using neural networks for automatic feature extraction should be straightforward and is left as future work.

## A.8 HYPERPARAMETER SETTINGS

We divide the full set of hyperparameters that our algorithm depends on into two groups: those that are common to all algorithms that use DDPG (Table 2), and those that are specific to skill chaining (Table 3). We did not try to optimize over the space of DDPG hyperparameters, and used the ones used in previous work (Lillicrap et al., 2015; Fujimoto et al., 2018). Table 3 shows the hyperparameters that we chose on the different tasks considered in this paper. Most of them are concerned with learning initiation set classifiers, the difficulty of which varies based on domain. To determine the correct setting of these parameters, we usually visualized the learned initiation set classifiers during the course of training (like Figures 2 and 6), and made adjustments accordingly.

| Parameter | Value |
|---|---|
| Replay buffer size | $1e6$ |
| Batch size | 64 |
| $\gamma$ | 0.99 |
| $\tau$ | 0.01 |
| Number of hidden layers | 2 |
| Hidden size 1 | 400 |
| Hidden size 2 | 300 |
| Critic learning rate | $1e-3$ |
| Actor learning rate | $1e-4$ |

Table 2: DDPG Hyperparameters

| Parameter | Point Maze | Four Rooms | Reacher | Ant Maze | E-Maze |
|---|---|---|---|---|---|
| Gestation Period ($N$) | 5 | 10 | 5 | 1 | 5 |
| Initiation Period | 1 | 10 | 3 | 0 | 1 |
| Buffer Length ($K$) | 20 | 20 | 20 | 750 | 20 |
| Option Max Time Steps ($T$) | 100 | 150 | 150 | 100 | 100 |

Table 3: Deep Skill Chaining Hyperparameters

### A.9    COMPUTE INFRASTRUCTURE

We used 1 NVIDIA GeForce 2080 Ti, 2 NVIDIA GeForce 2070 Ti and 2 Tesla K80s on the Google Cloud compute infrastructure to perform all experiments reported in this paper.

### A.10    NOTE ON COMPUTATION TIME

Each option is parameterized by its own neural networks, which are only updated when the agent is inside that option's initiation set. For a given transition, this leads to at most two or three updates. In Point-Maze, updating all options on a transition took $0.004 \pm 0.0003$ s more than just updating the global DDPG agent (averaged over 300 episodes using 1 NVIDIA 2080 Ti GPU) - a trivial amount of extra computation time.

