# OpenReview forum: "Option Discovery using Deep Skill Chaining"
_ICLR.cc/2020/Conference — Accept (Poster)_

### Official Review · AnonReviewer1 · 2019-10-20
**Official Blind Review #1**

**Rating:** 6

**Review:**

This paper studies the problem of learning suitable action abstractions (i.e., options or skills) that can be composed hierarchically to solve control tasks. The starting point for the paper is the (classic) observation that one skill should end where another can start. The paper then proposes a recursive algorithm for learning skills that obey this property. After finding some number of trajectories that reach the goal, the last few states of these trajectories are taken to define the initiation set for the ultimate skill and the termination set for the penultimate skill. The procedure is repeated, yielding a sequence (a "chain") of skills that extends from the initial state distribution to the terminal state. The fact that the number of skills is not defined apriori seems to be a strength, and the extension to trees of skills is neat.

The paper compares the proposed algorithm against state-of-the-art hierarchical baselines on five continuous control tasks of varying complexity; the proposed method outperforms baselines on 2 - 4 of the 5 tasks (depending on the setting considered).

Perhaps the biggest limitation of the paper is that it ignores the exploration problem. As noted in "Why Does Hierarchy (Sometimes) Work So Well in Reinforcement Learning?", exploration is empirically one of the main benefits of HRL, yet the proposed method, by construction, cannot take advantage of this strength.

I am leaning towards accepting this paper. While the idea is quite simple, it seems to work well empirically and, to the best of my knowledge, is novel. The simplicity should also make it a good launching point for future work in HRL. My main concern with the paper is that the poor performance of baselines on some of the tasks. For example, HIRO [Nachum 18] quickly solves AntMaze, while the baseline used in Fig 1 (HAC, which the paper claims is stronger than HIRO) gails to solve this task. I would be more confident about this paper if it added a comparison with HIRO.

Minor comments:
* "fixed and costly hyperparameter" -- Why is the number of skills "costly"?
* "Learning backward from the goal" -- Two other closely related papers are Policy Search by Dynamic Programming [Bagnell 04] and "Forward-Backward Reinforcement Learning" [Edwards 18]
* "it may choose to ignore learned options" -- Why should we expect that the optimal policy will use any skills, if it can be represented entirely using primitive actions?
* [Jinnai 2019] -- This citation is repeated in the references.

----------------- UPDATE AFTER AUTHOR RESPONSE --------------------------
Thanks for the discussion. I think this paper makes a useful contribution to the field, and would be a good ICLR paper. My reason for not voting for "accept" (rather than "weak accept") is that the experiments are not the most complex, and are all navigation-esque (reacher is effectively a point mass). Could we use this method to solve tasks of significantly greater complexity than those tackled in prior work?

**Experience Assessment:**

I have published one or two papers in this area.

**Review Assessment: Checking Correctness Of Derivations And Theory:**

I assessed the sensibility of the derivations and theory.

**Review Assessment: Checking Correctness Of Experiments:**

I assessed the sensibility of the experiments.

**Review Assessment: Thoroughness In Paper Reading:**

I read the paper at least twice and used my best judgement in assessing the paper.

---

> ### Author Response · Authors · 2019-11-13
> **Response to Review # 1**
>
> Thank you for taking the time to review our paper and providing insightful critiques. We hope that we can effectively answer your questions.
>
> >> Perhaps the biggest limitation of the paper is that it ignores the exploration problem.
>
> The focus of this paper is to show that deep skill chaining (DSC) autonomously discovers skills that solve the MDP with low sample complexity and high reliability. Others, like the Eigen-Option framework [1], discover skills that optimize for exploration. Combining these two methods to scale to even harder problems is an open problem, but is separate from the core argument of this paper.
>
> >> I would be more confident about this paper if it added a comparison with HIRO.
>
> HIRO [2] relies on using a dense L2 distance based reward functions to train both levels of their feudal hierarchy. By contrast, HAC [3] leverages Hindsight Experience Replay (HER) [4] to work in the more general sparse-reward setting. Since 9 out of the 10 subplots in Figure 1 of our paper are results from learning in sparse reward settings, we chose to compare against HAC instead of HIRO. To see why we consider the sparse reward setting to be more general, consider the following two examples:
>
>     1. In the Four Rooms with a Lock and Key environment, the agent must arrive at the lock with the key already in its possession. In this problem, the importance and the location of the key is not known apriori, making it hard to design a meaningful “distance-to-goal” reward function.
>
>     2. Perhaps even more generally, an RL agent may not always have access to its (x, y) coordinates or even the (x, y) coordinates of the goal state. Such situations arise naturally when observations are in the form of images or when the feature representation includes raw sensory data such as that coming from a laser range finder. In all of these settings, it would not be straightforward to use the dense distance based reward function required by HIRO.
>
> To verify our hypothesis that HIRO would not learn effectively in sparse-reward settings, we used the HIRO author codebase and experimented with the reward function used to train on the Ant-Maze task. We found that when the task’s reward function (used to train the higher level of the hierarchy) was changed from a L2 distance based reward to a sparse one, HIRO’s success rate fell from ~ 90% (reported) to ~ 20% (after training on 12 million environment steps). Furthermore, when both high and low level reward functions were changed from dense to sparse rewards, which is the setting used in most of our experiments, HIRO was unable to able to get > 0% success rate. This experiment highlights that HIRO is most suitable for domains which permit a dense L2 distance based reward. Since we evaluated our algorithm in the more general sparse reward setting, we chose to compare against HAC.
>
> We will update the Related Work section of our paper to emphasize that we compare against HAC and not HIRO because HAC doesn’t require a dense reward function and thus works in the more general sparse reward setting.
>
> >> "fixed and costly hyperparameter" -- Why is the number of skills "costly"?
>
> We consider the number of skills to be a costly hyperparameter because (a) it is often unclear how many skills are sufficient to solve a given problem (which depends on the difficulty of the problem and the expressiveness of the used policy class) and (b) deep option discovery methods tend to be sensitive to the particular setting of this hyperparameter. As a result, this hyperparameter is often chosen via an expensive grid search procedure.
>
> >> "Learning backward from the goal" -- Two other closely related papers are Policy Search by Dynamic Programming [Bagnell 04] and "Forward-Backward Reinforcement Learning" [Edwards 18]
>
> Thank you, we will add references to these papers in the related work.
>
> >> "it may choose to ignore learned options" -- Why should we expect that the optimal policy will use any skills, if it can be represented entirely using primitive actions?
>
> Theoretically, given infinite interactions with the environment and an arbitrarily powerful function approximator, an RL agent can asymptotically find the optimal policy comprising primitive actions only. However, given a sufficiently hard problem and a finite amount of training time, the agent finds that taking options (discovered using deep skill chaining) reliably gets it closer to its goal. This higher reliability of success when using options is due to the fact that each option has to solve a strictly easier sub-problem and thus only has to represent its own local value function.
>
> [1] Machado, Marios C et al "A laplacian framework for option discovery in reinforcement learning" ICML 2017
> [2] Nachum, Ofir, et al. "Data-efficient hierarchical reinforcement learning" NeurIPS 2018
> [3] Levy, Andrew, et al. "Learning multi-level hierarchies with hindsight" ICLR 2019
> [4] Andrychowicz, Marcin, et al. "Hindsight experience replay" NeurIPS 2017

---

> > ### Comment · AnonReviewer1 · 2019-11-15
> > **Thanks for the clarifications!**
> >
> > Thanks for the detailed response! This alleviates most of my concerns with the paper.
> >
> > HIRO: Thanks for clarifying this, and for running the comparison with sparse rewards. I hadn't realized that HIRO performed poorly on sparse reward tasks.
> >
> > > "However, given a sufficiently hard problem and a finite amount of training time, the agent finds that taking options (discovered using deep skill chaining) reliably gets it closer to its goal."
> > Out of curiosity, did you actually observe this in practice? In my experience with these sorts of algorithms, I usually find the opposite, where the agent collapses to only using a single skill.

---

> > > ### Author Response · Authors · 2019-11-15
> > > **Higher reliability of options**
> > >
> > > Thank you for taking the time to read our response!
> > >
> > > >> Out of curiosity, did you actually observe this in practice? In my experience with these sorts of algorithms, I usually find the opposite, where the agent collapses to only using a single skill.
> > >
> > > Yes, we do observe that options are a more reliable way to get closer to the goal than simply taking primitive actions. We think that this is because of two reasons:
> > >
> > > (a) the goal-directed nature of our skill discovery procedure ensures that successful execution of each option in our chain bring us closer to the goal and
> > >
> > > (b) because we explicitly learn initiation sets, options do not have to learn representations for states that lie far away from its initiation set. This allows each option to learn very good solutions in their local portion of the state-space.
> > >
> > > We hope that we were able to effectively answer all your questions. Thank you again for your time and feedback.

---

### Official Review · AnonReviewer2 · 2019-10-23
**Official Blind Review #2**

**Rating:** 6

**Review:**

Summary of the paper
The authors tackle hierarchical RL and build upon the work of (Konidaris & Barto, 2009b) to derive a skill chaining algorithm with non-linear function approximations.
During execution, skill chaining is an option-based algorithm, ie., a master policy (the option policy in the paper) selects one of O options and then delegates the control the corresponding sub-policy that executes primitive actions until their termination condition.
The learning consists of learning the master policy, here a DQN algorithm, as well as the sub-policies, each of which is a separate instance of DDPG (meaning, they do not share weights).
The gist of the learning algorithm, DSC, is to proceed backward. Initially the goal state is is the default termination condition.  As new options are added to the pool, their terminal condition simply correspond to the initial condition of their subsequent option, which consist in a learnt classifier that acts as indicator function during the learning phase.

General remarks
The paper is overall clear and well written. The background and related work section is informative and well structured. The experiments are sound (and all averaged over 20 different seeds-except for one).

From a performance perspective, DSC is comparable to the state of the art most of the time, and even improving upon it on some experiments. However, I am not totally convinced that the learnt skills can be interpretable as clearly indicating different regions of the environment.
Looking at Fig 3 and the attached video for instance, the skills oscillate and seem unstable overall.

Also, the choice of N (the number of the rollouts during the gestation period) and K (the segmentation threshold) seem to be crucial and very much application dependent. It would have been useful to plot the performance of DSC for different values of of these hyperparameters in order to show the potential “flatness” of the error surface.

Finally, when the learning starts, the (only) option o_G must reach the goal state N times. Which means, the initial DDPG agent has to gather a certain amount of successful rollouts.

- Isn’t that a strong condition, given that initially, hierarchical RL is meant to overcome long sequences issues in “flat” RL?
- Could you please elaborate on the training time of DSC?
- Given that DDPG fails at the Point E-Maze environment, how could DSC still learn new options?

Suggestions for improvement
- Page 4, in “learning the option-value function”: it is mentioned that the master policy can choose primitive actions. This part only becomes clear in Page 5 when it is mentioned that it happens through the global option.
- Page 4, in “adding new options…”, could you please clarify how the max of the returns is set as the initial Q-value. Since its’ not tabular, how can it be “assigned”? If I understood correctly, it is regressed towards this max return value, but I can’t find it clearly expressed in the text. But then, how does it affect the rest of the Q values?

Minor
1. T represents the transition function page 1 and is then overloaded to represent time steps (page 3) without notice.
2. page 5, paragraph 3.5: redundancy in “is is”

**Experience Assessment:**

I have read many papers in this area.

**Review Assessment: Checking Correctness Of Derivations And Theory:**

I carefully checked the derivations and theory.

**Review Assessment: Checking Correctness Of Experiments:**

I carefully checked the experiments.

**Review Assessment: Thoroughness In Paper Reading:**

I read the paper thoroughly.

---

> ### Author Response · Authors · 2019-11-13
> **Response to Reviewer # 2**
>
> Thank you for reviewing our paper and for providing useful feedback.
>
> >> Looking at Fig 3 and the attached video for instance, the skills oscillate and seem unstable overall.
>
> In Figure 3c, we erroneously repeated the colors of some of the option trajectories. For instance, the blue and green skills seem to repeat. However, in reality, all discovered skills are distinct and specialize in different regions of the state-space. We will of course fix the coloring scheme in our visualizations to remedy this.
>
> >> When the learning starts, the (only) option o_G must reach the goal state N times. Isn’t that a strong condition, given that initially, hierarchical RL is meant to overcome long sequences issues in “flat” RL?
>
> Yes, you are correct; DSC does require that. However, despite its reliance on exploration and the global agent to generate a small number of successful rollouts, DSC still combats the difficulty of learning effective policies in long-horizon problems. It does so by breaking up the overall problem into a sequence of sub-problems, and delegating options to represent the solutions to each of those sub-problems. We hypothesize that this decomposition of the long-horizon problem into a sequence of shorter-horizon sub-problems is the reason DSC drastically outperforms a flat DDPG agent.
>
> >> Given that DDPG fails at the Point E-Maze environment, how could DSC still learn new options?
>
> The Point E-Maze subplot in Figure 1a shows that while DDPG performs poorly, it doesn’t always get the lowest possible reward of -1500. DDPG is therefore able to sometimes reach the goal. However, it is unable to capitalize on those few occasions to learn an effective policy in this environment. By contrast, as soon as DSC collects a handful of successful trajectories (N = 5 in this task), it creates its first option. Given that this option can reliably drive the agent to the goal from states in its initiation set, the global DDPG agent must now merely focus on getting the agent to the initiation set of that option - a substantially easier problem than the one it was originally faced with. This process continues and every time a new option is created, it lowers the representational burden on the global DDPG agent, allowing the agent to rely on locally valid options to solve the overall problem.
>
> >> The choice of N and K seem to be crucial. It would have been useful to plot the performance of DSC for different values of these hyperparameters.
>
> Thank you for this suggestion. We have added Section A.3.3 and Figure 8 to our paper, which analyzes the algorithm’s sensitivity to the aforementioned hyperparameters. Our experimental results suggest that DSC is fairly robust to a wide selection of these parameters.
>
> >> Could you please elaborate on the training time of DSC?
>
> Since DSC uses DDPG as its flat level policy class, its training time is similar to that of DDPG. Since each option’s neural networks are only updated when the agent is inside its initiation set, DSC leads to at most two or three more updates per transition than a flat DDPG agent. In Point-Maze, this merely amounts to 4 milli-seconds (+/- 0.3 ms) of extra training time per transition (averaged over 300 episodes using 1 NVIDIA 2080 Ti GPU). As a result, training DSC on Point-Maze for 300 episodes is about 10 minutes longer than training a flat DDPG agent.
>
> >> Could you please clarify how the max of the returns is set as the initial Q-value. Since its not tabular, how can it be “assigned”? How does it affect the rest of the Q values?
>
> The initial Q-value of the new option node is set to the max of the sampled Q-values. This is done by setting the bias value of the new node in the neural network parameterizing the policy over options. This approach ensures that the Q-value predictions corresponding to the other options remain unchanged. Thank you for pointing this out, we have now updated our paper with this clarification.

---

### Official Review · AnonReviewer3 · 2019-10-23
**Official Blind Review #3**

**Rating:** 6

**Review:**

Summary
The authors tackle the problem of skill discovery by skill chaining. In particular, the authors claim two key contributions over the state of the art in option discovery 1) learn initiation sets 2) do not need to specify the number of options and this is also learned. Skill discovery is formalized by skill chaining; wherein the skills are chained backward from a goal state, and in a way, such that termination of an option is the initiation of the option that precedes in its chain.

Interesting and useful aspects of this work are goal-based learning of where options should initiate (although clarification on goals would be crucial), the discussion of the optimality of the discovered solutions, scales up various ideas already proposed in Konidaris & Barto (2009b).

Skill discovery through skill chaining, in particular, has not been extensively explored in a deep RL setting and serves as a useful contribution.

Detailed comments:
The authors introduce DSC with an intuition based on a goal. However, it is never mentioned where this goal comes from. In sec 3, the authors describe an algorithm based on the presumed goal. It is not clear to me: the goal is at one instance defined as \beta_{o_i} := \mathcal{I}_{o_i - 1}, and then in the next few lines, it is said “goal state of the MDP or the initiation condition of a previously learned option”. Is it correct to say this is an algorithm (sec 3) for goal-based option discovery using DSC, where the goal is specified in the MDP?

Before going into the details of the architecture, it would be useful for the reader to have a formalism or a clear algorithm where at least the definition of what a goal constitutes here is clearly stated.

Intra-option policy: It is nice to see that the option’s internal policy is not driven by the task-specific reward and has its internal reward. In the sparse reward setting: how is the subgoal reward chosen? In dense reward: what kind of distance metric is used? Please provide details.

Policy over options: The foremost option constructions seems a bit weird: is there a single goal specified which helps determine the termination condition of the global option? What would happen if there are multiple goals in the MDP?

Initiation set classifier: This is an interesting approach. Although, it raises a natural question: “
We continue adding to our chain of options in this fashion until a learned initiation set classifier contains the start state of the MDP. “ What happens if this never happens, or is this process is guaranteed to converge?

Experiments: Initiation set visuals are nice and interpretable. Experiments are not very convincing: authors mostly demonstrate results on control tasks that are specific to navigation and could do a more rigorous analysis by considering different tasks such as visual domains, robot manipulation tasks.  In particular, authors should compare their method and discuss other skill chaining approaches such as [1,2,3].

[1] Shoeleh, Farzaneh, and Masoud Asadpour. "Graph-based skill acquisition and transfer learning for continuous reinforcement learning domains." Pattern Recognition Letters 87 (2017): 104-116.
[2] Metzen, Jan Hendrik, and Frank Kirchner. "Incremental learning of skill collections based on intrinsic motivation." Frontiers in neurorobotics 7 (2013): 11.
[3] Konidaris, George, et al. "Robot learning from demonstration by constructing skill trees." The International Journal of Robotics Research 31.3 (2012): 360-375.

My primary concern is the amount of engineering that is needed to get this to work. There are multiple steps which do not seem to be sequentially dependent on the success of the previous steps (eg: global option construction needs a goal, the next options are only discovered once the global actions has been constructed, there is an initiation period and internal components of an option are all learned through multiple algorithms DQN, SMDP Q learning, DDPG).

Overall:
+Break the assumption that all options are everywhere.
+Number of options per task is also learned and does not need to be specified a priori.
+Chains skills in a smart way - which could be very useful for lifelong learning, but the approach, as it is, is limited by the goal of the MDP (whatever that means in this context, state or a Reward).
+Options are driven by internal rewards
-The heavy machinery used in the core algorithm seems to be inspired by Konandiais 2009b. It would be very useful to discuss, clarify and contrast what is novel and what is borrowed from Konandiais 2009b as is.

**Experience Assessment:**

I have published one or two papers in this area.

**Review Assessment: Checking Correctness Of Derivations And Theory:**

I carefully checked the derivations and theory.

**Review Assessment: Checking Correctness Of Experiments:**

I carefully checked the experiments.

**Review Assessment: Thoroughness In Paper Reading:**

I read the paper thoroughly.

---

> ### Author Response · Authors · 2019-11-13
> **Response to Reviewer # 3**
>
>
> >> it would be useful for the reader to have a formalism or definition of what a goal constitutes
>
> Thank you - we will definitely add such a formalism up-front.
>
> As pointed out in the Background, this paper addresses the problem of solving MDPs that have clearly defined goal states. While we do not require that the goal states be known, we must be able to check if the agent has successfully triggered the goal state in the MDP. In other words, we assume that we have access to an indicator function which we can query to determine if we have entered the goal region of the MDP. Note that this is a milder assumption than that made by RL algorithms that learn goal-conditioned policies and take in a goal state as input during the learning process (eg UVFA, HER, HAC etc). We will update our paper to be more clear about where the goal state comes from.
>
> While in this paper we have limited our discussion to end-of-episode goal events, in general these target events could be any state that we would like our agent to get to (for e.g, intrinsically motivating events). One could create skill chains to target such events, and it would not require any change in the DSC algorithm itself.
>
> >> Intra-Option policy learning: In the sparse reward setting: how is the subgoal reward chosen? In dense reward: what kind of distance metric is used?
>
> The option’s internal reward function mirrors the structure of the task’s reward function. In the sparse reward setting, the option gets a step penalty of -1 and a terminating subgoal reward of 0 for entering its termination region.
>
> In the dense reward setting, we use the negative distance from the state to the parent option’s initiation classifier as the reward function. Since initiation classifiers are represented using parametric classifiers, computing the distance to the classifier’s decision boundary is straightforward and can be done using most popular machine learning frameworks. For instance, when using scikit-learn for option $o_i$ with parent $o_{i-1}$, this is implemented as follows:
>
> $R_{o_i}(s, a, s') = 0$ if $\beta_{o_{i}}(s')$ else $-\mathcal{I}_{o_{i-1}}$.decision_function($s’$)
>
> >> Is there a single goal specified which helps determine the termination condition of the global option? What would happen if there are multiple goals in the MDP?
>
> Yes, we assume that the MDP has a distinct goal region and the agent can check when it has successfully entered such a region. This goal indicator function, which is given, is defined to be the termination condition of the foremost option. The termination condition of subsequent options is defined to be the initiation classifiers of parent options learned using the procedure outlined in Section 3.3. We will add this clarification to our paper.
>
> If there are multiple distinct goal states in the MDP (with corresponding indicator functions), each goal state would yield its own skill chain. In this case, the policy over options would learn to trigger the goal state that would yield the most cumulative (discounted) rewards. This would happen without any changes to the algorithm - the logic is similar to the skill tree discussion in our paper.
>
> >> “We continue adding to our chain of options in this fashion until a learned initiation set classifier contains the start state of the MDP.” What happens if this never happens, or is this process is guaranteed to converge?
>
> Leemon Baird [4] showed that even in very benign MDPs, an RL agent using function approximation may diverge. In the event that the global DDPG agent diverges, the deep skill chaining procedure might not be able to chain skills all the way back to the start state of the MDP. Intuitively however, the skill chaining procedure creates options that lie on the solution path of the MDP and thus can lower the probability that the policy will diverge. In all our experiments, DSC was able to chain skills back to the start state(s) of the MDP.
>
> >> In particular, authors should compare their method and discuss other skill chaining approaches such as [1,2,3].
>
> [1] and [2] reduce continuous state MDPs into a discrete graphs using state transitions sampled under a random walk. Skill discovery then proceeds on the discrete graph representation of the MDP. This approach of constructing a state-transition graph is vulnerable to the curse of dimensionality and hence does not scale easily to problems with high-dimensional observations.
>
> In [3], Konidaris et al rely on optimal, or near-optimal, demonstration trajectories to learn skills. This requires an expert human or an optimal solution to the whole problem before any options are learned. DSC has no such requirement. Furthermore, while they can solve relatively low-dimensional problems, our paper shows how the skill chaining framework can be extended to high-dimensional state and action spaces.
>
> References:
> [1, 2, 3] - same as review
> [4] Baird, Leemon "Residual algorithms: Reinforcement learning with function approximation." MLP 1995.

---

> > ### Comment · AnonReviewer3 · 2019-11-14
> > **Add clarifications to paper**
> >
> > Thank you for the clarifications.
> >
> > I would encourage the authors to add these clarifications in the final version of the paper.
> >
> > Re "we assume that the MDP has a distinct goal region and the agent can check when it has successfully entered such a region." Could you please elaborate on this region? Since we are in the function approximation setting, how do you specify this region?
> >
> > Re "There are multiple steps which do not seem to be sequentially dependent on the success of the previous steps (eg: global option construction needs a goal, the next options are only discovered once the global actions has been constructed, there is an initiation period and internal components of an option are all learned through multiple algorithms DQN, SMDP Q learning, DDPG)." I am curious to understand how straightforward these pieces are when put together? Are there situations when the algorithm fails?

---

> > > ### Author Response · Authors · 2019-11-14
> > > **Clarification on goals and integration challenges**
> > >
> > > Thank you for getting back to us so promptly! Based on your observations, we have already added some clarifications to our paper. We are still adding others and hope to have them all in soon.
> > >
> > > >> Could you please elaborate on this [goal] region? Since we are in the function approximation setting, how do you specify this region?
> > >
> > > We assume that the ability to check if the task’s goal has been met is provided by the environment itself. This is common in most benchmark goal-oriented RL tasks. For instance, in the benchmark maze-navigation tasks considered in our paper (introduced in [1]), the simulator gives a terminating positive reward when the agent is some small threshold away from the goal. The control tasks in gym [2] (eg, Mountain Car, Acrobot) also behave in a similar way. In Deep Mind Control Suite [3], even when the environment provides pixel observations, it notifies the agent when it triggers the goal of the task. Internally, DM Control suite does this by keeping track of the relevant dimensions of the current state and comparing it to goal state specified in the task itself (under tolerances also defined internally in the simulator).
> > >
> > > It is also worth noting that skill discovery methods such as HIRO and HAC make a stronger assumption than this because (a) they take the goal state as input to their learning algorithm, which in problems with pixel observations would correspond to having images of what the goal of the MDP looks like and (b) they require the programmer to design a function that determines whether the lower levels of the hierarchy have achieved their sub-goals. By contrast, DSC learns classifiers for termination conditions which it can query to determine if each option’s sub-goal condition has been met.
> > >
> > > >> I am curious to understand how straightforward these pieces are when put together? Are there situations when the algorithm fails?
> > >
> > > You are right, DSC does bring together several individual learning components. However, by casting each component of the framework into standard formulations, we are able to leverage several well studied areas of machine learning to implement the deep skill chaining idea. For eg, by casting the option selection problem as a Q-learning problem, we are able to use an off-the-shelf DQN to learn the policy over options.
> > >
> > > Early in the development process, it also seemed to us that integrating so many learning components could result in unstable training. However, in practice we found that breaking a problem into simpler sub-problems allows us to consistently find good policies in challenging problems. Even if we end up learning “bad options” in our skill tree, the policy over options will ensure that we simply revert back to the global DDPG agent in that portion of the state-space. Roughly speaking, this loosely lower bounds DSC’s performance with that of a flat DDPG agent.
> > >
> > > We have also shared our software implementation of deep skill chaining. Through the use of the simple_rl library [4], we have focused on creating a codebase that is simple, readable and easy to build on. We hope that our paper and our codebase will provide an easy starting point for research in option discovery using the skill chaining framework.
> > >
> > > Failure points of our algorithm: as pointed out in our paper, DSC would be unable to create a skill chain in problems in which the deployed exploration algorithm is unable to generate a small number of trajectories that reach the goal. But as long as we get those handful number of successful trajectories, DSC will successively make the problem easier.
> > >
> > > [1] Duan, Yan, et al. "Benchmarking deep reinforcement learning for continuous control." International Conference on Machine Learning. 2016.
> > > [2] Brockman, Greg, et al. "Openai gym." arXiv preprint arXiv:1606.01540 (2016).
> > > [3] Tassa, Yuval, et al. "Deepmind control suite." arXiv preprint arXiv:1801.00690 (2018).
> > > [4] Abel, David. “simple_rl: Reproducible Reinforcement Learning in Python”, ICLR Workshop on Reproducibility in Machine Learning, 2019.

---

### Public Comment · ~Tom_Zahavy2 · 2020-04-27
**Missing refernce**

Hello,
Very nice work!

We want to refer the authors to our paper "A Deep Hierarchical Approach to Lifelong Learning in Minecraft" https://arxiv.org/pdf/1604.07255.pdf"
To the best of our knowledge, this was the first work that learned a policy over options with a DQN. While we did not discover the options in this paper, but predefined them, a reference still seems appropriate as the mechanism for learning the policy over options is identical.

Best, the authors

---

### Decision · Program_Chairs · 2019-12-19

**Decision:**

Accept (Poster)

**Comment:**

This paper tackles the problem of autonomous skill discovery by recursively chaining skills backwards from the goal in a deep learning setting, taking the initial conditions of one skill to be the goal of the previous one. The approach is evaluated on several domains and compared against other state of the art algorithms.

This is clearly a novel and interesting paper. Two minor outstanding issues are that the domains are all related to navigation, and it would be interesting to see the approach on other domains, and that the method involves a fair bit of engineering in piecing different methods together. Regardless, this paper should be accepted.